# PCR diagnostics are insufficient for the detection of Diarrhoeagenic *Escherichia coli* in Ibadan, Nigeria

**Olabisi C. Akinlabi**[1], **Rotimi A. Dada**[2], **El-shama Q. A. Nwoko**[1], **Iruka N. Okeke**[1]*

**1** Faculty of Pharmacy, Department of Pharmaceutical Microbiology, University of Ibadan, Oyo, Nigeria,
**2** Faculty of Pharmacy, Bowen University Iwo and Department of Pharmaceutical Microbiology, College of Health Sciences, Medical Laboratory Science Programme, Ahmadu Bello University, Zaria, Nigeria

* iruka.n.okeke@gmail.com

**Data Availability Statement:** Whole genome sequence data were submitted to ENA and are available Genbank https://www.ncbi.nlm.nih.gov/genbank/ as Bioproject PRJEB8667.

## Abstract

Understanding the contribution of different diarrhoeagenic *Escherichia coli* pathotypes to disease burden is critical to mapping risk and informing vaccine development. Targeting select virulence genes by PCR is the diagnostic approach of choice in high-burden, least-resourced African settings. We compared the performance of a commonly-used multiplex protocol to whole genome sequencing (WGS). PCR was applied to 3,815 *E. coli* isolates from 120 children with diarrhoea and 357 healthy controls. Three or more isolates per specimen were also Illumina-sequenced. Following quality assurance, ARIBA and Virulencefinder database were used to identify virulence targets. Root cause analysis of deviant PCR results was performed by examining target sensitivity using BLAST, Sanger sequencing false-positive amplicons, and identifying lineages prone to false-positivity using in-silico multilocus sequence typing and a Single Nucleotide Polymorphism phylogeny constructed using IQTree. The sensitivity and positive predictive value of PCR compared to WGS ranged from 0–77.8% while specificity ranged from 74.5–94.7% for different pathotypes. WGS identified more enteroaggregative *E. coli* (EAEC), fewer enterotoxigenic *E. coli* (ETEC) and none of the Shiga toxin-producing *E. coli* detected by PCR, painting a considerably different epidemiological picture. Use of the CVD432 target resulted in EAEC under-detection, and enteropathogenic *E. coli eae* primers mismatched more recently described intimin alleles common in our setting. False positive ETEC were over-represented among West Africa-predominant ST8746 complex strains. PCR precision varies with pathogen genome so primers optimized for use in one part of the world may have noticeably lower sensitivity and specificity in settings where different pathogen lineages predominate.

## Introduction

Diarrhoea is a principal contributor to morbidity and mortality in young children, particularly in Africa and parts of Asia [1]. A range of pathogens can produce diarrhoea, including diarrhoeagenic *Escherichia coli* (DEC). *E. coli* is the most common proteobacterial intestinal

**Funding:** This work was supported by an African Research Leader's Award to INO jointly funded by the UK Medical Research Council (MRC) and the UK Department for International Development (DFID) under the MRC/DFID Concordat agreement and is also part of the EDCTP2 programme supported by the European Union (Award # MR/L00464X/1). This funding included full and partial doctoral stipend support for OCA and RAD respectively. INO is a Calestous Juma Science Leadership Fellow supported by the Bill and Melinda Gates Foundation (INV-036234). The funders had no role in study design, data collection and analysis, decision to publish, or preparation of the manuscript.

**Competing interests:** The authors have declared that no competing interests exist.

commensal and this presents a challenge for delineating potentially pathogenic strains, which harbor virulence genes, from those that are beneficial [2]. There are several DEC pathotypes, five of which have been validated in human volunteer challenge studies and/or significant outbreaks: enterotoxigenic *E. coli* (ETEC), enteropathogenic *E. coli* (EPEC), Shiga toxin-producing *E. coli* (STEC) (which includes enterohaemorrhagic *E. coli* (EHEC)), enteroinvasive *E. coli* (EIEC) and enteroaggregative *E. coli* (EAEC) [2]. Diffusely-adherent *E. coli* (DAEC), cell-detaching *E. coli* (CDEC) and cytolethal distending toxin-producing *E. coli* (CLDTEC) are among pathotypes that have been proposed but require more evidence to validate them. Within each pathotype, there is considerable genetic diversity and different pathotypes evolved convergently in separate lineages. There are also hybrid strains that carry virulence genes corresponding to multiple pathotypes [3–8].

Traditionally, DEC pathotypes identification uses tissue culture or vertebrate animal-based diagnostics, which are onerous, highly specialized, and therefore unsuitable for routine clinical use [9]. The Gold Standard for adherent DEC pathotypes (EPEC, EAEC, DAEC and CDEC) has been a tedious human epithelial adherence assay [2], which few labs can perform, while ETEC, STEC and CLDTEC Gold Standards depend on detection of the toxins that these pathotypes produce. EIEC can be identified by detecting entry into epithelial cells or conjunctivitis produced in rodent models [10]. Today, the genes encoding defining phenotypes for each pathotype are known, making it possible to detect DEC using DNA-based methods that offer greater throughput, more widespread application and animal-free diagnostics [11–13]. PCR for DEC in particular is a low-cost method that can leverage instrumentation and expertise initially acquired for other purposes and therefore present in most reference and research laboratories, and in some smaller clinical laboratories. While automated PCR-based protocols are rapidly gaining ground in high-income settings, the low cost for manual PCR set up, particularly for end-point PCR, and the flexibility around its consumable procurement mean that it remains an approach of choice in low-income settings, including Nigeria where we work. Because of the large number of targets needed to cover the most prominent DEC pathotypes, and the need to integrate DEC diagnosis into a larger search for diarrhoeal pathogens, multiplex PCR protocols are favored, even though these are notoriously challenging to optimize [14–16]. PCR is also perceived to have a lower false-positive rate than nucleotide hybridization, which was popular in the twentieth century but requires somewhat costly, and often inaccessible, detection methods. Moreover, hybridization can pick up similar but not identical genes and therefore confound DEC diagnoses [17,18].

The range of known virulence gene alleles and pathogenic lineages, as well as the absence of a precise genetic definition for EAEC and newer pathotypes, mean that the more targets that can be detected, the more sensitive a testing method becomes. For all of these reasons, whole genome sequencing offers the promise of reliably detection any DEC pathotypes, without the need of several cumbersome assays [19–21]. Due to the high cost and limited availability of WGS [22], most labs continue to use PCR-based tests that were developed and validated in a very limited number of settings against small panels of strains. We compared the performance of a popular PCR-based protocol that has been in use for almost two decades to WGS. We then identified root causes of PCR under-performance in our setting thereby uncovering more general principles for PCR diagnostics failure for pathogens that show considerable diversity.

## Materials and methods

### Study population and ethical considerations

This study was part of a larger investigation of the epidemiology of diarrhoeagenic *E. coli* isolated from 120 children with diarrhoea and 357 children without diarrhoea presenting to

primary health clinics in southwestern Nigeria with ethical approval from the University of Ibadan/University College Hospital ethics committee (UI/EC/15/0093). Patient parents or guardians provided written informed consent to participate. According to the approved protocol, neither PCR nor WGS results, which were performed for surveillance and not obtained in a timely manner, were used to inform patient care.

## Review of multiplex PCR methods

We systematically searched Pubmed https://pubmed.ncbi.nlm.nih.gov for "diarrhoeagenic *Escherichia coli*" and "PCR method". Abstracts retrieved were read to determine whether they included development of a protocol, and if so, where the protocol was developed. Where this was the case, the number of papers citing the reference in question and the location of cited studies was retrieved using Google Scholar https://scholar.google.com.

## Strains

Strains used in this study were collected as part of an epidemiological survey for diarrhoeagenic *E. coli* [7]. Up to ten isolates were obtained per stool specimen and all isolates were subjected to PCR screening. Up to three isolates showing different biochemical and/or antimicrobial susceptibility profiles were whole genome-sequenced. Control strains, which are well-studied and genome-sequenced DEC from other parts of the world, are listed in Table 1.

## Virulence Gene Detection by PCR

Molecular diagnosis of DEC was performed by gene amplification using PCR. The *E. coli* genomic DNA was screened by multiplex PCR by targeting the genes as described by Aranda *et al*., 2004 [14] (Table 1). However, the protocol was modified due to the closeness of the band sizes in Aranda's PCR2 targets of *est* and *stx2* genes, with band sizes 180 and 190 bp, respectively [23]. The specific modification splits PCR2 two separate reactions, designated PCR3 and PCR 4 (Table 2). Likewise, because of previously observed non-specific products, the annealing temperature of the PCR reactions was increased and the annealing time was reduced for each reaction to avoid previously detected non-specific products [23]. The

**Table 1. PCR oligonucleotides from the Aranda et al [14] protocol that were used in this study.**

| Primer Designation | Function encoded by target gene | Primers (5' to 3') | Target gene or probe | Amplicon size (bp) | Positive control strains |
|---|---|---|---|---|---|
| *eae f*<br>*eae r* | Intimin of EHEC and EPEC | CTGAACGGCGATTACGCGAA<br>CCAGACGATACGATCCAG | *eae* | 917 | E2348/69 |
| *bfp f*<br>*bfp r* | Bundle-forming pilus of EPEC | AATGGTGCTTGCGCTTGCTGC<br>GCCGCTTTATCCAACCTGGTA | *bfpA* | 326 | E2348/69 |
| *CVD432 f*<br>*CVD432 r* | Dispersin transporter | CTGGCGAAAGACTGTATCAT<br>CAATGTATAGAAATCCGCTGTT | *aatA* (CVD432) | 630 | 042,17–2,60A |
| *LT f*<br>*LT r* | Heat liable toxins of ETEC | GGCGACAGATTATACCGTGC<br>CGGTCTCTATATTCCCTGTT | *elt* (LT) | 450 | H10407 |
| *ST f*<br>*ST r* | Heat stable toxins of ETEC | ATTTTTMTTTCTGTATTRTCTT<br>CACCCGGTACARGCAGGATT | *est* (ST) | 190 | H10407 |
| *ipaH f*<br>*ipaH r* | invasion-associated locus of EIEC and *Shigella* | GTTCCTTGACCGCCTTTCCGATACCGTC<br>GCCGGTCAGCCACCCTCTGAGAGTAC | *ipaH* | 600 | E137 |
| *stx1 f*<br>*stx1 r* | Shiga toxin 1 and 2 of EHEC and STEC | ATAAATCGCCATTCGTTGACTAC<br>AGAACGCCCACTGAGATCATC | *stx1* | 244 | EDL933 |
| *stx2 f*<br>*stx2 r* | Shiga toxin 1 and 2 of EHEC and STEC | GGCATGTCTGAAACTGCTCC<br>TCGCCAGTTATCTGACATTCTG | *stx2* | 190 | EDL933 |

**Table 2. Polymerase Chain Reaction Cycle.**

| Step | PCR 1 Aranda et al, 2004 [14] | PCR 2 Aranda et al 2004 [14] | PCR 3 Aranda et al 2004 [14], Odetoyin et al 2016 [23] | PCR 4 Aranda et al 2004 [14] Odetoyin et al 2016 [23] |
|---|---|---|---|---|
| Target genes | *aatD*, *eae* and *bfpA* | *elt*, *est*, *ipaH*, *stx1* and *stx2* | *elt*, *est* and *ipaH* | *stx1* and *stx2* |
| 1. Hot start | 94˚C 3min | 94˚C 3min | 94˚C 3min | 94˚C 3min |
| 2. Denaturing | 94˚C 30sec | 94˚C 45sec | 94˚C 45sec | 94˚C 45sec |
| 3. Annealing | 60˚C 30 sec | 51˚C 30 sec | 51˚C 30 sec | 58˚C 30 sec |
| 4 Extension | 72˚C 1 min | 72˚C 45 sec | 72˚C 45 sec | 72˚C 45 sec |
| 5. Cycling | Steps 2–4 repeated 40 Times | Steps 2–4 repeated 40 times | Steps 2–4 repeated 40 times | Steps 2–4 repeated 40 times |
| 6 Terminal extension | 72˚C 7 min | 72˚C 7 min | 72˚C 7 min | 72˚C 7 min |

Amplicons were visualized on TAE agarose gels containing 1% agarose for PCR 1 and PCR 3 and 1.5% for PCR 4. The volume of agarose used provided 4 mm depth in the electrophoresis tray. Gels were visualized and photographed using the UVP GelMax Imager which utilizes a built-in midrange 302 nm ultraviolet (UV) transilluminator.

conditions used are as outlined in Table 2. All multiplex PCR was performed using the *illustra PuReTaq Ready*-To-*Go PCR Beads (GE Healthcare UK.)*. These contain *Taq* polymerase, BSA, stabilizers, dNTPs, 1,5nM MgCl$_2$, 50mM KCl and reaction buffer. Each bead is reconstituted to 25 μL by adding 2.5 μl of each forward and reverse primer solution, 20 ng/μl of *E. coli* DNA and nuclease-free water to complete the final volume. Well known prototypical strains were used as positive controls (Table 1) and DH5α and CFT073 as negative controls in the PCR reactions.

## Whole genome sequencing

*E. coli* genomic DNA were extracted using the Wizard Genomic Extraction kit (Promega) according to the manufacturer's instructions and they were shipped and bulk sequenced at the Wellcome Trust Sanger Institute using the illumina platform. The quality of raw reads were checked using FastQC and individual results were aggregated using MultiQC. ARIBA and Virulencefinder database, were used to identify virulence genes. Intimin allele calling was performed by comparing database prototypes to assembled genomes using BLAST. Sequence data were submitted to ENA and are available from ENA https://www.ebi.ac.uk/ena/browser/home and Genbank https://www.ncbi.nlm.nih.gov/genbank/ as Bioproject PRJEB8667.

## PCR amplicon sequencing

Selected isolates that amplified the virulence genes by PCR but did not harbor such genes by WGS were reamplified and amplicons were Sanger sequenced at Inqaba Biotec (Ibadan, Nigeria and Pretoria, South Africa). Returned sequences yielding single peaks were compared to the NCBI nucleotide database using BLAST to retrieve the match and flanking sequences. Primer sequences were then compared for within amplicon/ flanking sequence matches.

## Mapping PCR diagnostic results onto phylogenies

All sequenced isolate reads were mapped to EAEC_042 reference genome (accession number: FN554766) to create a multi-fasta alignment file. Using the multi-fasta alignment file, Single Nucleotide Polymorphism (SNP) were called using SNP-sites and the SNP file was used to

construct a phylogenetic tree using IQTree software [24]. Microreact [25] was used to visualize phylogenetic trees, STs and loci identified by PCR and WGS.

## Quantitative evaluation of PCR performance

The performance of PCR for detecting each pathotype was compared to WGS using standard methods [26]. Percentage sensitivity was determined by dividing the number of the true positives (TP) by the sum of number of the true positives (TP) and the numbers of false negatives (FN) then multiplied by 100 [TP/(TP+FN) * 100]. Percentage specificity was determined by dividing the number of the true negatives (TN) by the sum of numbers of the true negatives (TN) and the number of false positives (FP) then multiplied by 100 [TN/(TN+FP) * 100]. The positive predictive value was determined by dividing the number of true positives with the (TP) by the sum of the number of true positive (TP) and the number of false positives (FP) then multiplied by 100 [TP/(TP+FP) * 100]. The negative predictive (NP) value was determined by dividing the number of true negatives (TN) with the sum of the number of true negatives (TN) and the number of false positives (FN) then multiplied by 100 [TN/(TN+FN) * 100].

## Statistical analysis

Epi Info version 7 software (Centers for Disease Control and Prevention, Atlanta, GA, USA) was used for statistical calculations. Fisher's exact test and Chi square test were used to test the association of test-defined pathotypes with disease. P-values < 0.05 were considered significant, with Bonferroni corrections used where applicable.

## Results

### Multiplex PCR for DEC is rarely developed in Africa but commonly used to identify DEC in African settings

A search for "diarrhoeagenic *Escherichia coli*" and "PCR method" on Medline (Pubmed) initially yielded 207 hits, which when reviewed uncovered 16 studies that developed multiplex PCR protocols for diarrhoeagenic *E. coli* (Table 1). In contrast to some other diagnostic protocols, where one or a few methods gain prominence over time, there are a wide variety of published PCR protocols for DEC with 7 of those published before 2010 having 1,132 or more citations (Table 3). A similar number of studies developed after 2010 are accumulating citations as well. Five protocols were developed and published from Europe, four protocols were developed by groups working in Asian countries and three by South American groups. Only two of the protocols were published by groups based in Africa (both in Egypt) and none in Nigeria, West Africa or elsewhere in the WHO Africa (AFRO) region (Table 3). While target choice has been compared across different protocols [27], we were not able to find studies that included a validation on isolates from the WHO-AFRO region nor any studies comparing different PCR protocols with one another.

　　Within Africa, we found clear geographic bias in protocol selection that could not, from the aforementioned, be attributed to validation, among studies seeking DEC in human, food or environmental specimens (Table 3). Some well-cited protocols [27,29,36,39] have rarely or never been used in Africa. The Vidal et al 2005 [28] protocol has been used in the most African countries but not more than two or three times in any single country. This study evaluated the Aranda et al (2004) [14] protocol, which uses a minimal primer set to identify all five pathotypes that have been validated through human volunteer studies and/or outbreaks. This was the most commonly employed protocol in Nigeria and some of its primer pairs are employed

**Table 3. Application of published DEC PCR identification protocols on the African continent.**

| Protocol (Reference) | Developed in | #Google Scholar citations as at 1 Dec 2022 | # Citations from African use study | Location of African studies citing the protocol* |
|---|---|---|---|---|
| Toma et al 2003 [15] | Japan | 297 | 11 | Cote d'Ivoire, Kenya, Nigeria, Tanzania, South Africa, Uganda |
| Aranda et al 2004 [14] | Brazil | 294 | 18 | Ghana, Kenya, Nigeria, South Africa |
| Vidal et al 2005 [28] | Chile | 301 | 24 | Bukina Faso, Chad, Ethiopia, Egypt, Ghana, Kenya, Nigeria, South Africa and multi-country |
| Muller et al 2007 [29] | Germany | 56 | 0 | |
| Metwally et al 2007 [30] | Egypt | 22 | 2 | Egypt, Nigeria |
| Antikainen et al 2009 [9] | Finland | 93 | 15 | Bukina Faso, Chad, Malawi, South Africa |
| Tobias et al 2012 [27] | Sweden | 69 | 2 | Ghana, South Africa |
| Hegde et al 2012 [31] | India | 70 | 4 | Burkina Faso, Ethiopia, Nigeria, Sudan |
| Fialho et al 2013 [32] | Brazil | 12 | 0 | |
| Helmy et al 2013 [33] | Egypt | 7 | 1 | Nigeria |
| Al Talib 2014 [34] | Malaysia | 18 | 0 | |
| Oh et al 2014 [35] | Korea | 22 | 0 | |
| Onori et al 2014 [36] | Italy | 80 | 3 | Ghana, South Africa, Sudan |
| Naziri et al 2015 [37] | Iran | 0 | 0 | |
| Sjoling et al 2015 [38] | Sweden | 37 | 15 | Burkina Faso, Cote D'Ivoire, Ghana, Kenya, South Africa |
| Zhang et al 2020 [39] | China | 15 | 0 | |

in other popular multiplex protocols [27,32,40]. The protocol originally presented by Aranda et al (2004) [14] consists of two multiplex PCRs that detected a total of eight targets, all validated DEC pathotypes. Noting that the amplicon sizes for *stx1* (180 bp) *elt* (190 bp), we previously separated PCR2 into two reactions to avoid errors of interpretation [41].

## The epidemiological picture for the same specimen set is radically different when DEC are identified by PCR and WGS

The Aranda et al (2004) [14] PCR performed well for all positive and negative control strains, which were included in every batch, reproducibly giving the expected result every time. *E. coli* from 413 specimens obtained from children in Ibadan, totaling 3,815 isolates and averaging eight isolates per specimen, were tested by PCR. All PCR positives were sequenced and in instances where there were <2 such positives from any specimen, three isolates with different biochemical profiles were selected for sequencing. Overall, quality assurance of the whole genome sequences revealed that 99.97% of the reads were of suitable quality for downstream analyses, with a mean quality score (Phred score) of at least 30 across different base positions. Reads also had per base N content of less than 8%. The median N50 of assembled genomes was 225,560bp and the range 381bp—1,307,997bp. Due to resource limitation, a smaller number of 1,230 (averaging three per specimen) were tested by whole genome sequencing but all

**Table 4. DEC pathotypes identified by WGS and PCR from the case and control specimens.**

| Pathotypes | Cases (%) n = 120 | | Controls (%) (n = 357) | | Total specimens (n = 477) | | P-values | |
|---|---|---|---|---|---|---|---|---|
| | **PCR** | **WGS** | **PCR** | **WGS** | **PCR** | **WGS** | **PCR** | **WGS** |
| **EAEC** (CVD432) | **25(20.8)** | **35(29.2)** | **74(20.7)** | **102(28.6)** | **99** | **137** | 0.9804 | **0.9075** |
| **EHEC/STEC** | **9(7.5)** | **0(0)** | **39(10.9)** | **0(0)** | **46** | **0** | 0.2807 | **0** |
| *stx2* | 8(6.7) | 0(0) | 35(9.8) | 0(0) | 43 | 0 | 0.2992 | 0 |
| *stx1* | 1(0.8) | 0(0) | 1(0.2) | 0(0) | 2 | 0 | 0.4176 | 0 |
| *stx1, stx2* | 1(0.8) | 0(0) | 3(0.8) | 0(0) | 4 | 0 | 1 | 0 |
| **EPEC** | **15(12.5)** | **7(5.8)** | **28(8.4)** | **16(4.5)** | **43** | **23** | 0.1233 | 1.3202 |
| *eae+ bfp+* | 1(0.8) | 1(0.8) | 3(0.8) | 2(0.6) | 4 | 3 | 1 | 1 |
| *eae* | 1(0.8) | **7(5.8)** | 12(3.4) | **16(4.5)** | 13 | 23 | 0.2001 | 0.5499 |
| *bfp* | 6(5) | 0(0) | 13(3.6) | 0(0) | 19 | 0 | 0.5103 | 0 |
| **ETEC** | **41(34.2)** | **8(6.7)** | **109(30.5)** | **11(3.1)** | **150** | **19** | 0.4582 | **0.0823** |
| *elt+est+* | 12(10) | 1(0.8) | 15(4.2) | 0(0) | 27 | 1 | **0.0174** | 0.2516 |
| *elt* | 26(21.7) | 7(5.8) | 49(13.7) | 11(3.1) | 75 | 8 | **0.0387** | 0.1711 |
| *est* | 5(4.2) | 2(1.7) | 10(2.8) | 0(0) | 15 | 2 | 0.5446 | 0.0629 |
| EIEC/Shigella (*ipaH*) | 9 | 0(0) | 22(6.2) | 1(0.3) | 31 | 1 | 0.6071 | 0 |

pathotypes identified by PCR were sequenced for validation. As shown in Table 4, in spite of this overlap, the epidemiological picture revealed by PCR is very different from that defined by WGS.

Table 4 shows that PCR detected all five pathotypes and identified ETEC 150 (31.4%) as the most common pathotype (31 *est*-positive strains, 206 *elt*-positive strains and 67 strains having both targets), followed by CVD432-positive EAEC 99 (20.8%), STEC positive *stx1* and/or *stx2* 46(9.6%), *eae* and/or *bfp*-positive EPEC 43 (9%) and EIEC from which *ipaH* amplified 33 (6.9%). Thus PCR detected all five DEC pathotypes as shown in Fig 1 and ETEC with *est* and *elt* genes or *elt* alone were significantly associated with diarrhoea (p< 0.05), based on PCR results. The *stx1* and *stx2* genes were detected in 12 and 63 isolates, respectively, by PCR. WGS did not verify any of the *E. coli* strains as harboring a Shiga toxin gene. Thus, PCR identified 70 isolates that could be classified as STEC but none of the PCR-identified 'STEC' were verified by whole genome sequencing. No EHEC (*eae*-positive STEC) were identified by either method.

We compared sensitivity and specificity of PCR to WGS for each of the pathotypes. The CVD432 (*aatA*) gene was targeted for EAEC, *eae* and *bfp* for EPEC, *elt* and *est* for ETEC, *ipaH* for EIEC, *stx1* and *stx2* for STEC, Table 5 shows that PCR sensitivity for DEC identification was 77.8% for ETEC, 61.7% for EAEC, 45% for EPEC and 0% for both EIEC and STEC. PCR did show better specificity for EAEC (91.6%), EPEC (94.7%), EIEC (93.7%), STEC (90.1%). ETEC specificity was considerably lower (74.5%) (Table 5).

## Low sensitivity of EAEC detection by PCR is due to too few targets

The Aranda et al (2004) protocol, like most other DEC multiplex protocols, seeks only one EAEC target, CVD432. Whole genome sequencing on the other hand allows the investigator to screen strains for a total of 14 loci associated with EAEC [7,42], allowing for greater sensitivity and accounting for the extremely low sensitivity of PCR. When we compared the prevalence of CVD432 detected by PCR to that from WGS among isolates that were tested for both. PCR identified 260 (21%) EAEC isolates while WGS identified 355 (28.7%).

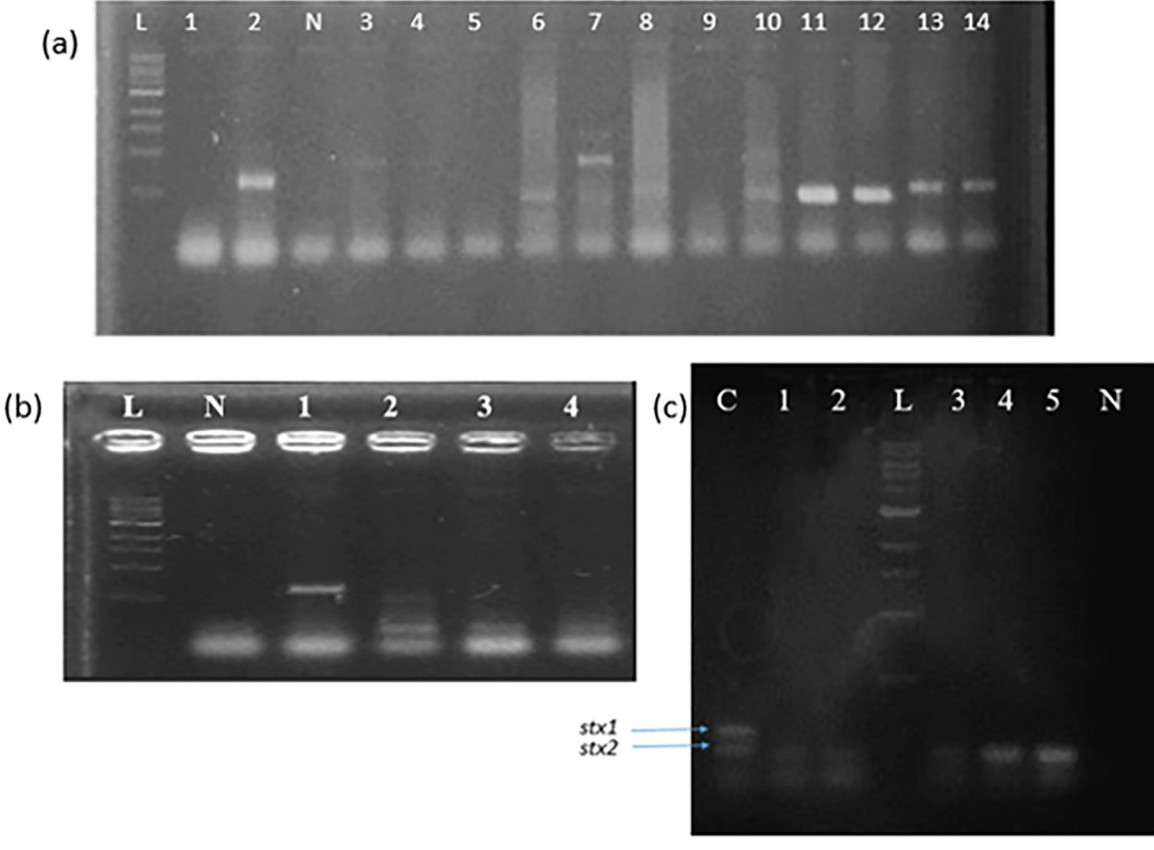

**Fig 1.** A selection of multiplex PCR gels (a) Multiplex PCR1 with lane L, 1-kb size ladder, Lane 2, EAEC 042 reference control, Lane N, negative control, lane 3–4 *E. coli* strains INOLLD97B and INOLLH029A positive for *eae* gene, lane 6, 11, 12, positive isolated *E. coli* strains with *bfp* gene, Lane 7, isolated *E. coli* strains with the *bfp* and *eae* genes, lane 10, EPEC E2348/69 reference control with the *bfp* and *eae* genes, lane 13 and 14 *E. coli* strains INOLWH056C and INOLLH297C positive for CVD432 gene (b) Multiplex PCR3 with lane L, 1-kb size ladder, Lane N, negative control, Lane 1, EIEC E137 reference control with *ipaH* gene, lane 2 ETEC H10407 reference control with the *elt* and *est* genes, lane 3–4 isolates INOMNH199C and INOLLD018D *E. coli* strains positive for *est* gene. (c) Multiplex PCR4 with EHEC EDL933 reference controls (Lane C). Lane 1–5, strains INOMND61D, INOLLH045A, INOLLH030F, INOMNH077E and INOLKH217A with stx2 genes, Lane L, 1-kb size ladder.

## Reduced specificity for *bfpA* PCR and reduced sensitivity of *eae* primers

A prominent red flag in the PCR results was the 32 strains that repeatedly tested positive for *bfpA* but were *eae* negative (Fig 1). Typical EPEC strains harbor the chromosomal locus of enterocyte effacement (LEE) marked by *eae* and the EPEC adherence factor (EAF) plasmid-borne *bfp* genes whilst atypical EPEC have *eae* on the LEE without *bfp* and the EAF plasmid. Strains carrying *bfp* without *eae* are rarely reported. In this study, all 40 isolates from which

**Table 5. Evaluation of PCR performance for identification of diarrhoeagenic *E. coli* compared with sequence data.**

|  | EAEC | ETEC | EPEC | EIEC | STEC |
|---|---|---|---|---|---|
| % Sensitivity | 61.7 | 77.8 | 45.0 | 0.0 | ND* |
| % Specificity | 91.6 | 74.5 | 94.7 | 93.7 | 90.1 |
| Positive predictive value (%) | 73.9 | 10.7 | 27.3 | 0.0 | ND* |
| Negative predictive value (%) | 86.1 | 98.8 | 97.5 | 99.8 | 100 |

*There were no STEC true positives.

**Table 6. Intimin alleles of LEE-positive EPEC strains not detected by PCR.**

| Strain | Intimin allele | Prototype accession number | Sequence of *eae1* and *eae2* priming sites (mis-matched bases are not in bold) |
|---|---|---|---|
| E2348/69 (control) | alpha (28) | AF022236.1 | **CTGAACGGCGATTACGCGAA**...880 nt...**CTGGATCGTATCGTCTGG** |
| LKD064A | rho | DQ523613.1 | **CTGAACGGCGATTACGC**AAA...880 nt...**CT**GGA**TCGTATCGTCTGG** |
| LLH22C | alpha | AF022236.1 | **CTGAACGGCGATTACGCGAA**...880 nt...**CTGGATCGTATCGTCTGG** |
| LLH35B | alpha | AF022236.1 | **CTGAACGGCGATTACGCGAA**...880 nt...**CTGGATCGTATCGTCTGG** |
| LLH012D | kappa | KT591304.1 | **CTGAACGGCGATTACGCGAA**...880 nt...**CTGGATCGTATCGTCTGG** |
| LLD100C | iota2 | AB647371.1 | **CTGAACGGCGATTACGCAA**...880 nt...**CTGGATCGTATCGTCTGG** |
| LLH332A, LLH332B, LLH332E | episilon | AB647573.1 | **CTGAACGGCGATTACGC**GA...880 nt...**CTGGATCGTATCGTCTGG** |
| LL197, LLH161I, LLH161F | lambda | AJ715409.1 | **CTGAACGGCGATTACGCGAA**...880 nt...**CTGGATCGTATCGTCTGG** |
| LLH336B, LLH336E, LLD104C, MNH109F, LLHO33E, LLH261C1 | theta2 | FM872418.1 | **TG**C**ACGGCGATTACGCGAA**...880 nt...**CTGGATCGTATCGTCTGG** |

*bfpA* was amplified did not have the gene as determined by WGS. This includes seven *eae*-positive strains that were misclassified as EPEC by PCR. A total of 19 atypical EPEC strains (*eae*-positive and *bfpA*-negative) were not identified by PCR. WGS and screening by VirulenceFinder revealed that 17 of these strains carried all other LEE markers. We therefore hypothesized that they may carry intimin alleles that are not amplified by the *eae* primer pair. We used the *eae* gene from prototypical EPEC strain E2348/69 to retrieve *eae* from each of the assembled genomes of these strains using BLAST. BLAST was then used to type each intimin allele against database prototypes (Table 6). We additionally examined each allele for complementarity to the *eae1* and *eae2* primers. The results of these analyses are summarized in Table 6 and show that while seven of the strains sequenced carried intimin alpha, kappa and lambda alleles with no mismatches in the primer annealing sites, and therefore should have amplified, ten strains carried the iota, rho and theta intimin alleles, which have between one and three mismatches with the primers, which have the possibility of reducing amplification at conditions optimal for other alleles. Of note is the fact that in addition to the Aranda et al [14] protocol, these *eae* primers also feature in multiplex PCRs for DEC by Nazari et al, Puno- Sarmiento et al, and Dias et al [13,43,44] and so this problem is likely to occur with some other multiplex protocols.

## False positivity with *elt* primers is due to non-specific amplification in unfamiliar lineages

In this study, false positives for most genes were scattered across the phylogeny (Fig 2). This was true to some extent for *elt*. However, more *elt* false-positivity was seen within phylogroup D and in particular in the ST720 and ST69 complexes complex. ST69 is a well-characterized ST complex that includes extraintestinal and enteroaggregative *E. coli* strains. False-positive ETEC were detected in ST 69 (10/31 isolates) and ST394 (4/12 isolates). The false positivity rate of 14/50 was significantly greater than the overall false positivity rate (p = 0.03). ST720 complex is less well studied and in this study comprised isolates belonging to ST 8746 (n = 51), ST 720 (n = 13), ST8749 (n = 8), ST8131 and ST1136 (4 each), with the highest number of isolates and greatest false-positive rates in ST8646 (17/51) and ST720 (6/13). Altogether 17/51 ST8746 strains gave a false-positive *elt* result, as compared to 213 (5.6%) overall (p = 0.0006) and the false-positivity rate in ST720 (6/13) was even greater. Of the 17 ST8746 complex strains that were erroneously identified as ETEC by PCR, all but three were in fact EAEC

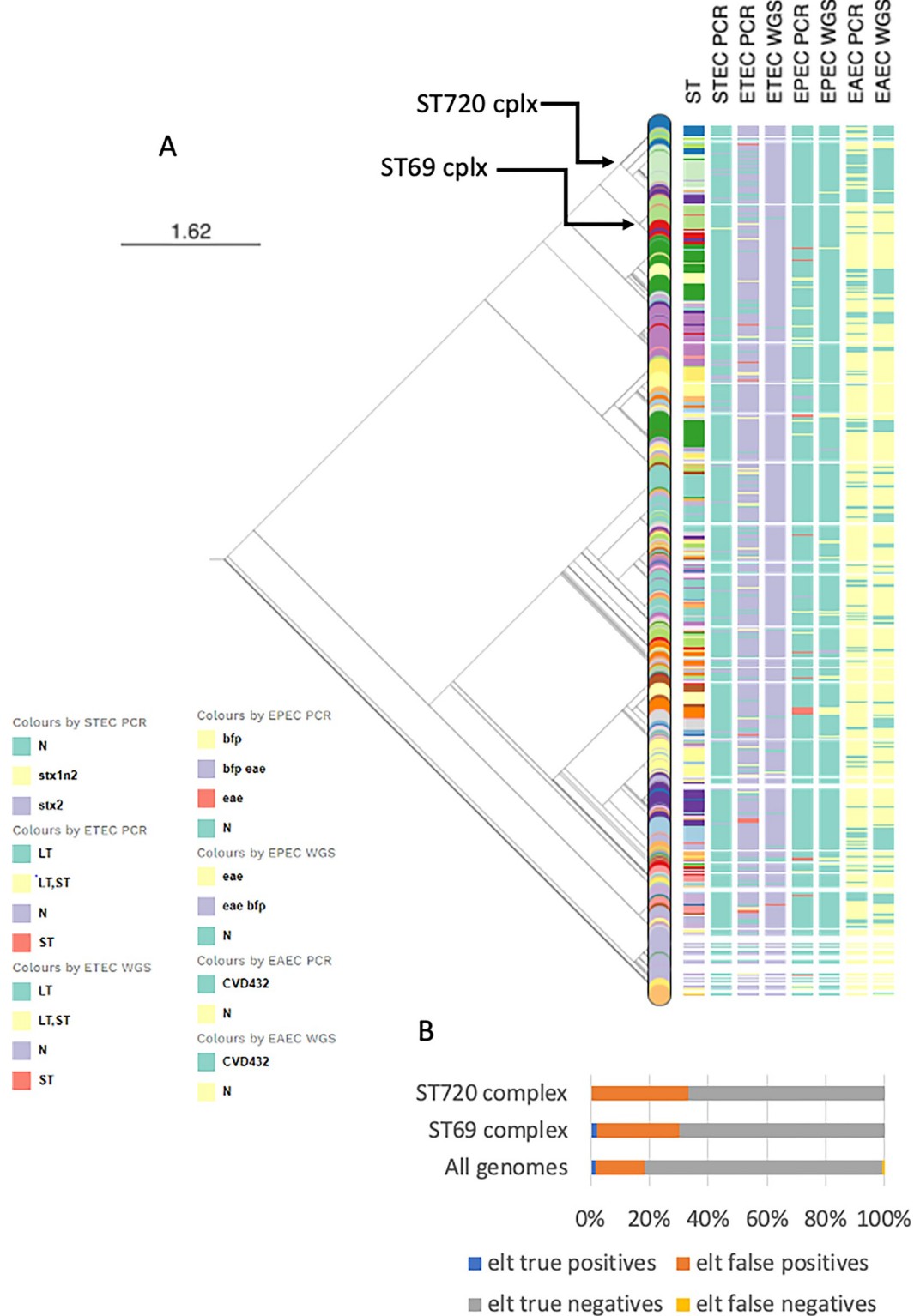

**Fig 2. Maximum likelihood SNP tree of genomes used in this evaluation showing sequence type, results for *stx1*, *stx2*, *stx1+2* PCR positivity; *elt*, *est* and *elt+est* PCR and WGS positivity, and CVD432 PCR and WGS positivity.** The figure can be engaged interactively at https://microreact.org/project/pWbLpY3qhV6Npa9q8Rpgrk-wgs-n-pcr. While there was some concordance for CVD432, there was little for the STEC and ETEC toxins. Nodes for the ST720 and ST69 complexes in which *elt* false positives are over-represented are indicated with arrows. The matrix to the left of the tree is colored for

positivity to specific specific targets indicated in the figure key or negativity (N) for those targets. (b) False and true positives, and negatives, for *elt* overall and for ST720 and ST69 complexes.

strains (Fig 2). Thus, the PCR method for this same lineage simultaneously overestimated the prevalence of ETEC, underestimated EAEC and hid from existence a previously unreported EAEC lineages. STs are numbered chronologically with STs in the Wirth et al (2006) [45] Achtman *E. coli* scheme above 1500 recorded after the year 2010. Thus, most STs within the complex, including ST8746, are relatively new to science. Of interest is that ST8746 itself was the fourth commonest ST we detected in this study, showing that it is very common in our setting. One strain in the complex, an ST720 strain from Germany, had been recorded in the database as at 2010. The relatively recent registration of other isolates in the ST complex is likely due to the very few sequence-based studies that take place in Nigeria and nearly locales. Other ST8746 strains have been reported from Ghana (n = 1) and United Kingdom (n = 1) [46,47] but the largest existing collection is from our own study. The data point to the likelihood that false positives with the *elt* primers will be common in Nigeria and other settings where ST8746 organisms predominate but likely less so in areas where this lineage is absent or less prominent. They also illustrate the weakness of PCR methods primarily used in West Africa that are not developed here.

**The *stx* multiplex produces a confounding non-specific amplicon due to mis-priming.** In previous studies, we have de-complexed Aranda et al (2004)'s second multiplex, which was originally designed to contain five primer pairs [23]. Thus, we use one multiplex to detect the *elt*, *est* and *ipaH* genes for ETEC and EIEC and a second duplex PCR for *stx1* and *stx2*. The *stx* duplex PCR reaction identified 12 *stx1*- and 63 *stx2*-positive strains. Upon WGS analysis, none of these isolates, nor any of the other isolates sequenced in the study carried these genes. We sought to understand why PCR yielded a right-sized product in the *stx* duplex and attempted Sanger sequencing of a few amplicons, most of which failed due to multiple peaks suggesting that each band consisted of multiple amplicons. We did get a product for the *stx* duplex PCR4 that was presumed to be the right size for *stx1* from two strains. Flanking a 192 bp region of the amplified sequence are low complexity regions (Fig 3) with good partial matches for the *stx1*-forward and *stx2*-reverse primers providing a rational explanation of how a '190 bp' amplicon could be produced in a multiplex reaction. As shown in Fig 3, the matching sequence itself represents part of the gene encoding a DNA injecting protein of a temperate bacteriophage Gally Phage LF82_763 (Accession number OV696608), for which several *E. coli* in the database are lysogenized.

## Discussion

### Multiplex PCR performance for DEC identification is poor

Diarrhoeagenic *E. coli* (DEC) are endemic in Africa where a wide range of lineages produce life-threatening infections in young children [48,49]. The majority of clinical and research laboratories in West Africa are unable to detect these pathogens, therefore available evidence of their importance comes largely from short-term epidemiological surveys. Of those labs that do diagnose DEC infections, or perform surveillance, multiplex PCR is typically the technique of choice with multiple protocols published in the literature. PCR remains popular for DEC identification here and elsewhere in the world and most of the available information on DEC epidemiology comes from studies employing different PCR methodologies [9,14–16,27,28,39–41,50–56], PCR precision may vary with pathogen genome so that primers optimized for use in one part of the world may not have sufficient precision in geographies where different

**Fig 3.** (a) stx mis-priming amplicon sequence with matches >6 contiguous base pairs on any stx primer highlighted. (b) Corresponding highlighted regions in stx primers, indicated in the same colors.

pathogen lineages predominate. However, there does not appear to be evidence that any have been optimized for use in West Africa where the pathogen lineage repertoire may differ from the rest of the world.

In this study, we found that the Aranda et al (2004) [14], which performed well in its initial assessment on isolates from Brazil, has been used extensively in Nigeria, including by us, and has limited sensitivity and specificity. This is one of several multiplex PCR protocols, some of which have been well validated, but not on local African pathogenic lineages. It also includes primer pairs for EAEC, ETEC and EPEC that are used in other multiplex protocols. In our 2016–2019 Ibadan, Nigeria setting, the Aranda et al (2004) [14] protocol under-estimated the prevalence of EAEC to some degree because of lowered sensitivity of the CVD432 primer for their target and to a much larger degree because too few EAEC targets are included in the multiplex. The ETEC pathotype is well-defined with highly conserved virulence targets, the *elt* and *est* genes. It is also a vaccine priority and there is therefore considerable interest in understanding ETEC epidemiology in lesser surveilled parts of the world, including Nigeria. PCR overestimated the prevalence of ETEC and inferred an association with disease because of nonspecificity of ETEC primers and resulted in a locally predominant EAEC lineage being misclassified as ETEC. PCR led to misidentification of strains belonging to other pathotypes or commensals as Shiga toxin-producing *E. coli*, which are reportable, again because of primer nonspecificity. Typical EPEC numbers were also inflated, and atypical EPEC numbers diminished due to mis-priming. Overall PCR methodology offers reasonable sensitivity or specificity, but not both, for identifying some DEC. This has implications for the use of this method in clinical diagnosis and for interpreting the results of epidemiological surveys [12]. As PCR deficits are misinforming on specific lineages (such as ST720 complex) or strains carrying specific virulence gene alleles (e.g. intimin theta), the overall importance of these clades or factors is likely to be overlooked and could be significant. For example, intimin theta is among newer alleles reputed to confer enhanced invasion on atypical EPEC [57].

That PCR is not as sensitive as WGS is not a surprise. However, when the departure in results that PCR generates is widely deviant from reality, the value of PCR testing must be questioned. We note with interest that under-performance of PCR is attributable to different root causes for the different pathotype targets. The insensitivity diagnostic protocols that target CVD432 alone is a generalized problem with the pathotype and its current definition. The CVD432 locus, now known as *aatD* encodes part of a secretion system for the antiaggregation protein, Aap, and is present in only some EAEC. It is common among EAEC strains from some locations, for example the Chile and India strain sets on which that the "EAEC probe" Baudry et al (1990) was validated [58]. However subsequent data has shown that CVD432-negative EAEC strains predominate in some geographies, including Nigeria [7,59]. The insensitivity of CVD432-based PCR protocols and could be resolved, at least in part by using additional targets as Czeczulin et al and Boisen et al [60,61] have recommended. Including more primer pairs will add to the cost of detecting EAEC and, if they are multiplexed, could add risk of mis-identifications unless rigorous validation is performed.

Sensitivity was even lower for EPEC than EAEC, although EPEC are less common overall. In this case, the number of targets cannot be faulted since EPEC strains that have other LEE genes typically have *eae*. However, we find primer mismatches in some EPEC intimin alleles that were described after the original report of intimin allelic variation by Adu-Bobie et al (1998) [28]. While the divergence of these alleles from other intimins has been reported previously [62], potential mis-priming at consensus *eae* priming sites has not.

Unlike EAEC, for which the absence of a precise genetic definition requires multiple targets to be used to identify strains from this pathotype [7,42], ETEC and STEC have genetic definitions based on the presence of toxin genes that encode the virulence factors producing the disease. ETEC are vaccine priority organisms and STEC are reportable. Therefore, respective ETEC and STEC specificities in this study of 74.5% and 90.1%, respectively, are below what is needed to inform interventions and contain these pathogens.

The major advantage of whole genome sequencing is that most DEC pathotypes can be predicted with a high degree of accuracy from sequence data [20]. The obvious deterrent for routine use is cost. When isolate-based investigation of pathogens in diarrhoea stools is conducted, multiple genomes will have to be sequenced per specimen, thus WGS DEC studies are particularly expensive. It is important for resource-constrained labs in endemic areas to be able to pick up key pathogens and enabling this while sacrificing some sensitivity and specificity is justifiable. Imperfect diagnostics can be useful for identification of sporadic and outbreak cases and, to some degree for epidemiologic assessments. However, it is clear that some benchmarking or validation is necessary before reliable conclusions can be drawn from DEC PCR. Some popular protocols have been validated on less than 10 strains per pathotype [28], which is far lower than the number of lineages known at the time the test was developed, a number that continues to grow. DEC identification by PCR is inexpensive and popular in resource limited settings in Africa. However, an established DEC multiplex PCR protocol is under-performing in our setting. Based on the results of this study, we would argue that the complete lack of specificity for the vaccine prioritized pathotype ETEC and misidentification of STEC and EHEC, which are reportable foodborne pathogens make this methodology, by itself, cost-inefficient. Molecular microbiology methods that are performed simply because they can be, but are unable to achieve the aims for which they were developed outside of their originally developed context are rightly currently being phased out [63]. Unfortunately, there are few alternatives to end-point PCR-based DEC diagnostics that can be used broadly across West Africa, at least until WGS becomes more affordable and better disseminated. In the interim when existing multiplex PCR protocols are applied in settings different from those used to develop them, they can used alongside other methodology to verify outcome.

It is probable that more purposeful PCR protocol development and validation will yield more reliable diagnostics. Reducing multiplexy, designing new primers and testing them against genome sequences, using degenerate primers that can target multiple variants of each target, employing multiple controls from different lineages, and including housekeeping gene targets for quality assurance are all approaches that could ensure that PCR works better and problems, when they occur, are identified. These adaptations, as well as the careful testing that must be used to validate them will incur costs, which need to be measured against the falling costs of whole genome sequencing to determine cost effectiveness.

## Conclusion

Pathotype lineages vary with geographic location, thus PCR sensitivity may vary based on geographic depending on the predominating pathotype lineage. Multiplex PCR is an inexpensive option for identifying DEC pathotypes but cannot be relied upon to perform optimally in settings where lineages different from those on which protocols are optimized for are predominant. Where possible, epidemiological surveys for DEC should be whole-genome sequence-. In the instances that resource constraints do not permit this additional validation steps will be required to ensure that target loci for DEC identification are appropriate and that presumptive DEC identified by PCR indeed harbor the target genes used to define them.

## Acknowledgments

We thank David A. Kwasi, Anderson O. Oaikhena, Taiwo Badejo, Anthony Underwood, Ayorinde Afolayan, Erkison Ewomazino Odih, Catherine Ladipo and A Oladipo Aboderin for technical assistance and helpful comments.

## Author Contributions

**Conceptualization:** Olabisi C. Akinlabi, Iruka N. Okeke.

**Data curation:** Olabisi C. Akinlabi, Rotimi A. Dada, El-shama Q. A. Nwoko.

**Formal analysis:** Olabisi C. Akinlabi, Rotimi A. Dada.

**Methodology:** Olabisi C. Akinlabi, El-shama Q. A. Nwoko, Iruka N. Okeke.

**Visualization:** Rotimi A. Dada, Iruka N. Okeke.

**Writing – original draft:** Olabisi C. Akinlabi, Iruka N. Okeke.

**Writing – review & editing:** Olabisi C. Akinlabi, Rotimi A. Dada, El-shama Q. A. Nwoko, Iruka N. Okeke.

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
