## [Decision Letter · Decision Letter 0]

3 Mar 2023

PGPH-D-22-02114

Insufficiency of PCR diagnostics for Detection of Diarrhoeagenic Escherichia coli in Ibadan, Nigeria

Dear Dr. Okeke,

Thank you for submitting your manuscript to PLOS Global Public Health. After careful consideration, we feel that it has merit but does not fully meet PLOS Global Public Health’s publication criteria as it currently stands. Therefore, we invite you to submit a revised version of the manuscript that addresses the points raised during the review process.

We look forward to receiving your revised manuscript.

Kind regards,

Cemil Kurekci

Academic Editor

Journal Requirements:

2. We do not publish any copyright or trademark symbols that usually accompany proprietary names, eg  ©, ®, ™  (e.g. next to drug or reagent names). Please remove all instances of trademark/copyright symbols throughout the text, including ® and ™ on page 10.

3. Please provide separate figure files in .tif or .eps format only and remove any figures embedded in your manuscript file. Please also ensure that all files are under our size limit of 10MB.

4. We have noticed that you have uploaded Supporting Information files, but you have not included a list of legends. Please add a full list of legends for your Supporting Information files after the references list. 

Additional Editor Comments (if provided):

Reviewers' comments:

Reviewer's Responses to Questions

**Comments to the Author**

1. Does this manuscript meet PLOS Global Public Health’s publication criteria? Is the manuscript technically sound, and do the data support the conclusions? The manuscript must describe methodologically and ethically rigorous research with conclusions that are appropriately drawn based on the data presented.

Reviewer #1: Yes

Reviewer #2: Yes

2. Has the statistical analysis been performed appropriately and rigorously?

Reviewer #1: Yes

Reviewer #2: Yes

3. Have the authors made all data underlying the findings in their manuscript fully available (please refer to the Data Availability Statement at the start of the manuscript PDF file)?

Reviewer #1: Yes

Reviewer #2: Yes

4. Is the manuscript presented in an intelligible fashion and written in standard English?

Reviewer #1: Yes

Reviewer #2: Yes

5. Review Comments to the Author

Reviewer #1: The study is well planned and conducted. However, there are some points that need to be clarified before publication. Need to rearrangement on the title. The conclusion should align with the objectives and this should be emphasized.

Reviewer #2: Manuscript ID- PGPH-D-22-02114

Comments:

Materials and Method

- Please provide details of study population. Although it has been mentioned in abstract and Supporting information, it should also be in main body (materials section) of the manuscript.

Results

- Line 202- ... E. coli from413 specimens obtained from children in Ibadan..- How many children (120??)

- Some sentences in result section which are explainig the reason for the research findings should be written in Discussion section.

For example- Some sentences in the paragraph/ under heading of line 274. So, You can keep sentences of findings in this result section and other that are explaining the reasons behind the resuts or findings from other studies can be kept in Discussion section.

Discussion

- Add "can" after "They" in the sentence of line 406 ... used to develop, they 'can' be used.

6. PLOS authors have the option to publish the peer review history of their article (what does this mean?). If published, this will include your full peer review and any attached files.

**Do you want your identity to be public for this peer review?** For information about this choice, including consent withdrawal, please see our Privacy Policy.

Reviewer #1: **Yes: **Dr. Tarana Jahan

assistant Professor of Microbiology

Reviewer #2: No

---

## [Decision Letter · Decision Letter 1]

22 May 2023

PGPH-D-22-02114R1

Underperformance of Multiplex PCR for Diarrhoeagenic Escherichia coli Pathotype Detection in Ibadan, Nigeria

Dear Dr. Okeke,

Thank you for submitting your manuscript to PLOS Global Public Health. After careful consideration, we feel that it has merit but does not fully meet PLOS Global Public Health’s publication criteria as it currently stands. Therefore, we invite you to submit a revised version of the manuscript that addresses the points raised during the review process.

Two independent reviewers have assessed the manuscript and have raised some concerns that should be addressed before it can be accepted for publication. Please pay special attention to the comments by reviewer three regarding the whole-genome sequencing results.

We look forward to receiving your revised manuscript.

Kind regards,

Chrispin Chaguza, Ph.D

Academic Editor

Journal Requirements:

Additional Editor Comments (if provided):

Reviewers' comments:

Reviewer's Responses to Questions

**Comments to the Author**

1. If the authors have adequately addressed your comments raised in a previous round of review and you feel that this manuscript is now acceptable for publication, you may indicate that here to bypass the “Comments to the Author” section, enter your conflict of interest statement in the “Confidential to Editor” section, and submit your "Accept" recommendation.

Reviewer #2: All comments have been addressed

Reviewer #3: (No Response)

2. Does this manuscript meet PLOS Global Public Health’s publication criteria? Is the manuscript technically sound, and do the data support the conclusions? The manuscript must describe methodologically and ethically rigorous research with conclusions that are appropriately drawn based on the data presented.

Reviewer #2: Yes

Reviewer #3: Yes

3. Has the statistical analysis been performed appropriately and rigorously?

Reviewer #2: Yes

Reviewer #3: Yes

4. Have the authors made all data underlying the findings in their manuscript fully available (please refer to the Data Availability Statement at the start of the manuscript PDF file)?

Reviewer #2: Yes

Reviewer #3: (No Response)

5. Is the manuscript presented in an intelligible fashion and written in standard English?

Reviewer #2: Yes

Reviewer #3: Yes

6. Review Comments to the Author

Reviewer #2: (No Response)

Reviewer #3: Major comments

1. The authors compare the performance of multiplex PCR to WGS. At the core of this is to demonstrate assay sensitivity and specificity. How did the authors determine assay sensitivity, what was the limit of detection for each of the singleplex PCR? Were there any loss in sensitivity when these singleplex PCCR assays were combined in a multiplex PCR? With regards to specificity, did the authors test each panel against closely related strains? Which microorganisms were tested

2. Did the authors include internal and extraction controls to assess whether negative PCR were indeed negative, which housekeeping genes were included. Was there a consideration to include degenerate primers?

3. Where are the WGS QC results

Minor

1.The manuscript is not well referenced. Background has a number of missing references.

2. Figure legends need some attention

7. PLOS authors have the option to publish the peer review history of their article (what does this mean?). If published, this will include your full peer review and any attached files.

**Do you want your identity to be public for this peer review?** For information about this choice, including consent withdrawal, please see our Privacy Policy.

Reviewer #2: No

Reviewer #3: No

---

## [Editor Report · Decision Letter 2]

11 Jul 2023

PCR diagnostics are insufficient for the detection of Diarrhoeagenic Escherichia coli in Ibadan, Nigeria

PGPH-D-22-02114R2

Dear %TITLE% Okeke,

We are pleased to inform you that your manuscript 'PCR diagnostics are insufficient for the detection of Diarrhoeagenic Escherichia coli in Ibadan, Nigeria' has been provisionally accepted for publication in PLOS Global Public Health.

Best regards,

Chrispin Chaguza, Ph.D

Academic Editor
